

# 1  An Efficient Algorithm for Improved Doppler Profile
# 2  Detection of MST Radar Signals

Nimmagadda Padmaja[1], Souri Varadarajan[2], Polisetti Yashoda[3]  Enugonda Ramyakrishna [4]
[1]Professor, Department of Electronics and Communication Engineering, Sree Vidyanikethan Engineering
College, Tirupati, Andhra Pradesh, India-517102.
[2]Professor, Department of ECE, S V University College of Engineering, Tirupati, Andhra Pradesh, India.
[3]Scientist D, National Atmospheric Research Laboratory, Indian Space Research Organization, AP, India.
[4]Junior Research Fellow (ISRO Project), Sree Vidyanikethan Engineering College, Tirupati, AP, India.
*Correspondence to*: N.Padmaja (padmaja202@gmail.com)
**Abstract:** An efficient algorithm based on Empirical Mode Decomposition (EMD) de-noising using soft
threshold techniques for accurate doppler profile detection and Signal to Noise Ratio (SNR) improvement of
MST Radar Signals is discussed in this paper. Hilbert Huang Transform (HHT) is a time-frequency analysis
technique for processing radar echoes which constitutes EMD process that decomposes the non-stationary
signals into Intrinsic Mode Functions (IMFs).  HHT process has been applied on the time series data of MST
(Mesosphere-Stratosphere–Troposphere) radar collected from NARL (National Atmospheric research
Laboratory), Gadanki, India.  Further, spectral moments were estimated and signal parameters such as mean
doppler, signal power, noise power and SNR were calculated. Stacked doppler profile was plotted to observe the
improvement in doppler detection. It has been observed that there is a considerable improvement in recognition
of the doppler echo leading to improved Signal Power and SNR. The algorithm was tested for its efficacy on
various data sets for all the 6 beams and the results of two data sets are presented.
**Keywords:** MST Radar, Empirical Mode Decomposition, De-noising, Hilbert Huang Transform, Doppler, SNR
**1. Introduction**

23       Processing and analysis of radar echoes from MST region pose serious challenges to traditional signal
processing techniques especially at higher altitudes above 15 Kms since the echo returns are very weak and
buried in noise. The most common approach for analysis of atmospheric radar signals is the Fast Fourier
Transform (FFT) and Wavelets. However, Fourier Transforms are not suitable for applications that involve
nonlinear and non stationary signals as it has a serious drawback that in transforming to the frequency domain,
time information is hidden and not explicitly visible. Wavelet Transform analysis, which is widely used method,
overcomes some of the above limitations but there is a need for a-priori knowledge about the kind of scale
elements present in the signals and the choice of a suitable mother wavelet for analysis since the accuracy of the
results depends on the selection of the wavelet.  Hence, an adaptive signal processing technique based on the
Empirical Mode Decomposition (EMD) and de-noising using soft thresholding is proposed in this work.
**2. Indian MST Radar System**
The MST Radar facility at Gadanki (13.5° N, 79.2° E, 6.3° N mag.lat) is an excellent system used for
atmospheric probing in the regions of Mesosphere, Stratosphere and Troposphere (MST) covering up to a height
of about 100 km.  It is used for coherent backscatter study of the ionospheric irregularities above 100 Km. MST
radar is a state-of-the-art instrument capable of providing estimates of atmospheric parameters with very high



resolution on a continuous basis which contribute to study different dynamic process in the atmosphere. It is an important research tool in the investigation of prevailing winds, waves, turbulence and atmospheric stability and other phenomenon. The Indian MST radar is highly sensitive, pulse-coded, coherent VHF phased array radar operating at 53 MHz with a peak power-aperture product of $3 \times 10^{10}$ $Wm^2$.

## 3. Hilbert Huang Transform

Hilbert Huang Transform (HHT) is one of the best time-frequency analysis technique for processing radar echoes. HHT basically comprises Empirical Mode Decomposition process based on numerical shifting to decompose the non-stationary signal into Intrinsic Mode Functions and obtain instantaneous frequency solution. To obtain instantaneous frequency data, Hilbert Spectral Analysis (HSA) method is applied to the IMFs. Since the signal is decomposed in time domain and the IMFs length is the same as the original signal. HHT preserves the characteristics of the varying frequency. Also the, extraction of IMFs enables various de-noising techniques to be applied for accurate detection of doppler echo. This is the key benefit of HHT. It has been tested and validated comprehensively, but only empirically. In all cases studied, from any of the traditional analysis methods, HHT (Padmaja et al., 2011; Jing-tian et al., 2007) gave much sharper results in time-frequency-energy representations. HHT process was applied on time series MST radar data and was investigated for its efficacy. The spectral moments were estimated and signal parameters such as mean doppler, signal power, noise power and SNR were calculated.

### 3.1 Empirical Mode Decomposition

The Empirical Mode Decomposition is an effective de-noising technique (Flandrin et al., 2004; Padmaja et al., 2017) that can be used to process non-linear and non-stationary data. This method is adaptive and intuitive. Each oscillatory mode is represented by an IMF that satisfies the conditions of an IMF (Huang et al., 2005). An IMF represents a simple oscillatory mode and it is a counterpart to the simple harmonic function, but it is much general instead of constant amplitude and frequency, as in a simple harmonic component. IMF can have a variable frequency and amplitude as function of time.

### 3.1.1 EMD Algorithm

Given a non-stationary signal x(t), the EMD algorithm (Rilling et al., 2003) can be summarized as follows:

a) Find the local maxima and minima of the signal, then connect all the maxima and minima of signal $X(t)$ and obtain the upper envelope $X_u(t)$ and the lower envelope $X_l(t)$ respctively.

b) Compute the local mean value $m_1(t)=(X_u(t)+X_l(t))/2$ of data $X(t)$, subtract the mean value from signal $X(t)$ and get the difference : $h_1(t)=X(t)-m_1(t)$.

c) Assume $h_1(t)$ as new data and repeat steps(1) and (2) for k times, $h_1k(t)=h_1(k-1)(t)-m_1(t)$, where $m_1k(t)$ is the mean value of $h_1(k-1)(t)$ and $h_1k(t)$. Step(c) is terminated untill the resulting data satisfies the two conditions of an IMF, defined as $c_1(t)=h_1k$. The residual data $r_1(t)$ is expressed as $r_1(t)= X(t)-c_1(t)$.

d) Assume $r_1(t)$ as new data and repeat steps (a-c) and extract all the IMFs. Terminate the sifting process untill $n^{th}$ residue $r_n(t)$ becomes less than a predetermined number or the residue becomes monotonic.



e)   Repeat steps (a-d) till the residual no longer contains any useful frequency information. The original

signal is equal to the sum of its IMFs. If we have 'n' IMFs and a final residual $r_n$ (t), the original signal

X(t) can be defined as follows

$$X(t) = \sum_{i=1}^{n} Ci + rn \quad --------- (1)$$

**3.2 Intrinsic Mode Functions**
After the application of EMD (George Tsolis et al., 2011; Flandrin et al., 2004; Norden et al., 1998; Wu et al.,
2004; Dejie Yu et al., 2010), if the residue, $r_1$ still contains information of longer period components, then it is
again treated as new data and subjected to the sifting process. The sifting process can be stopped by any of the
predetermined criteria: either when the component value $c_n$ or the residue $r_n$ ,becomes less than the
predetermined value or also when the residue, $r_n$ becomes a monotonic function from which no more IMFs  can
be extracted. The criterion for  stopping the sifting process is based on limiting the size of the Standard
Deviation (SD) ,which can be computed from the two consecutive sifting results as shown in the equation (2).

$$SD = \sum_{t=0}^{T} \frac{|h1(k-1)(t) - h1k(t)|^2}{h1^2(k-1)(t)} \quad ---------- (2)$$

A typical value for SD is around 0.21 to 0.3. Hard and soft thresholding (similar to de-noising in Wavelets) is
used to treat Intrinsic Mode Functions to achieve high SNR.
**4. Denoising**
Signal de-noising scheme based a multiresolution approach is referred to as empirical mode decomposition de-
noising(Padmaja et al., 2017; Donoho et al, 1995). A smooth version of the input signal can be obtained by
thresholding the IMFs before signal reconstruction. Thresholding are of two types namely Hard and Soft
Threshold. If $\Gamma[\tau_j]$ is a thresholding function, and $\tau_j$ is the threshold parameter, the threshold can be determined
in different ways. Donoho and Johnstone proposed a universal threshold, $\tau_j$ for removing noise (Donoho et al,
1995). The method of soft threshold is applied to process the radar data. After extracting the Intrinsic Mode
functions in each range bin, de-noising techniques are employed before reconstruction of the Doppler spectra by
using threshold levels.
**4.1 Hard Threshold**
Hard threshold removes the corresponding IMFs depending on the frequencies if $\tau_j$ is less than or equal to 1. The
condition for hard threshold as shown in equation (3) and (4)
$f_j$ (t) =  $IMF_j$ (t)     If $|IMF_j$ (t)$| > \tau_j$

If $|IMF_j$ (t)$| \leq \tau_j$ -------------------------------------(3)

$f_j$ (t) =  $IMF_j$ (t)     If $|IMF_j$ (t)$| > \tau_j$

If $|IMF_j$ (t)$| \leq \tau_j$ -------------------------------------(4)

**4.2 Soft Thresholding**
Soft thresholding process tends to shrink noise towards zero. By taking the median values of IMFs,  $\tilde{\sigma}_j$  and  $\tau_j$
were calculated using equations (5), (6),and (7).
$\tau_j = \tilde{\sigma}_j$ sqrt(2. loge(N)) --------------------------------------(5)




$\sigma\tilde{}_j = MAD_j / 0.6745$ --------------------------------------(6)
$MAD_j = Median \{|IMF_j (t) - Median \{IMF_j (t)\}|\}$-------(7)
Where $\sigma\tilde{}_j$ is the estimation of the noise level of the $j^{th}$ IMF (scale level) and $MAD_j$ represents the absolute
median deviation of the $j^{th}$ IMF. The soft thresholding shrinks the IMF samples by $\tau_j$ towards zero as follows.
$\hat{f}_j (t) = IMF_j (t) - \tau_j$        If $IMF_j (t) \geq \tau_j$

0            If $|IMF_j (t)| < \tau_j$

$IMF_j (t) + \tau_j$        If $IMF_j (t) \leq -\tau_j$ -----------------(8)

The signal can be reconstructed by adding all the IMFs which gives the de-noised signal. This procedure is
applied for all the range bins. The processing steps are discussed in 4.21.

### 117    4.2.1 Processing steps for the Algorithm

(i)    Read the time series data ( in ' .r' format).

(ii)   Convert the .r file into .mat file and  read the data from .mat file.

(iii)  Calculate IMFs  by using EMD method for each range bin and apply Hilbert Transform on each IMF .

(iv)   Apply Soft Threshold technique of de-noising as mentioned below.

•    Calculate the noise level of the IMF viz. $\sigma\tilde{}_j$   (Donoho et al, 1995)  by finding the median values

of the IMFs.

•    Calculate the value of universal Threshold ($\tau_j$) by using the calculated value $\sigma\tilde{}_j$ .

•    Taking universal Threshold as reference and the conditions proposed by Donoho and Johnstone,

soft thresholding was done for each range bin.

•    If  the value of IMF is greater than or equal to $\tau_j$ , the IMF will be subtracted by $\tau_j$ . If the value of

IMF is less than $\tau_j$ then IMF value is made zero and if the value of IMF is less than or equal to - $\tau_j$

, then the IMF will be incremented by $\tau_j$..

•    This process is applied on all the range bins.

(v)   Reconstruct the signal by adding all the IMFs for each range bin and  apply three point running

average method  to  each range bin.

(vi)  Calculate the mean noise level for each range bin. (Hildebrand et al., 1974).

(vii) Subtract the mean noise level for each range bin and plot the stacked doppler spectrum.

(viii) Calculate the spectral moments viz., Total power (Zeroth moment), Doppler shift (First moment),

136            Spectral width (Second moment)

### 137    5. Moments Calculations

Three lower order Spectral moments (zero, first and second) and SNR   are calculated by using adaptive
moments method (Anandan et al., 2004). These three spectral moment represents the signal strength (power),
the weighted mean doppler shift and width of the spectrum (Woodman et al., 1985; Morse et al., 2002; Anandan
et al., 2004). The moments were calculated for the data of $24^{th}$  July 2002 and $22^{nd}$ Jan 2007 data by using FFT
and HHT.  The expressions for the first three moments are as follows.
The $0^{th}$ moment representing the total signal power is
$M_0 = \sum_{i=m}^{n} P_i$      --------------------------------------------(9)



The 1$^{st}$ moment representing the weighted mean Doppler shift is
$M_1 = (1/M_0) \sum\limits_{i=m}^{n} P_i f_i$  ------------------------------(10)
The 2$^{nd}$ moment representing the variance, a measure of dispersion from the mean frequency is
$M_2 = (1/M_0) \sum\limits_{i=m}^{n} P_i (f_i - M_1)^2$  -------------(11)
Where m, n are the lower and upper limits of the Doppler bin of the spectral window. $P_i$, $f_i$ are the powers and
frequencies corresponding to the Doppler bins within the spectral window.
Signal-to-noise ratio (SNR) in dB is calculated by equation (12).
$\mathrm{SNR} = 10\log\left(\dfrac{M_0}{N.L}\right)$  ------------------------------(12)
Where N and L are the total number of Doppler bins and mean noise level respectively which on multiplication
gives the total noise over the whole bandwidth.
Doppler width, which is taken to be the full width of the Doppler spectrum is calculated as:
Doppler Width $= 2\sqrt{(M_2)}$ ------------------------(13)
**6.Results And Discussion**
The developed HHT algorithm was applied on various range bins of time series MST Radar data upto 25 Kms.
The corresponding plots for different range bins are shown in figures 1-6.  It can be observed that developed
algorithm is able to identify the true peaks of the echo signal. The corresponding Mean Doppler profile plots
using both HHT and FFT are plotted in figures (7a) and (7b).  Spectral moments for East beam of 24$^{th}$ July 2002
and 22$^{nd}$ Jan 2007 data using FFT and HHT are plotted in figures 7a-8f. The average values of SNR, Power and
Noise for independent beams of two different data sets are tabulated in Table1. The average values of SNR,
Power and Noise for all the beams are tabulated in Table 2. It is clearly visible from the results that using the
proposed algorithm, the genuine doppler is detected accurately and also there is an improvement of 5.6587 dB in
SNR for 24 July 2002 data and 4.5667 dB improvement of SNR for 22 Jan 2007 data.
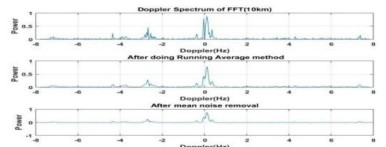  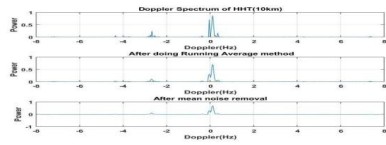
Figure 1:Doppler spectrum using FFT of height 10km     Figure 2:Doppler spectrum using HHT of height 10km

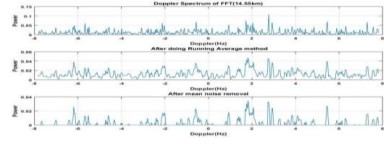  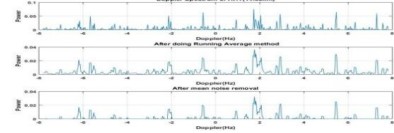


Figure 3:Doppler spectrum using FFT of height 14.55km     Figure 4:Doppler spectrum using HHT of height 19.5km






Figure 5:Doppler spectrum using FFT of height 20.25km    Figure 6:Doppler spectrum using HHT of height 20.25km

Fig(7a):Mean Doppler profile    Fig(7b): Mean Doppler profile    Fig(7c): Doppler plot    Fig(7d): Doppler plot
  of 24 July 2002 using FFT    of 24 July 2002 using HHT    extracted from Fig (7a)    extracted from Fig (7b)


Fig(7e): Power plot for    Fig(7f): SNR plot for    Fig(7g):Doppler width for    Fig(7h): Noise plot for
East beam of 24 July 2002    East beam of 24 July 2002    East beam of 24 July 2002    East beam of 24 July 2002


Fig(8a): Mean Doppler profile    Fig(8b): Mean Doppler profile    Fig(8c): Doppler plot    Fig(8d):Doppler plot
using FFT of 22 Jan 2007    using HHT of 22 Jan 2007    extracted from Fig (8a)    extracted from Fig (8b)

Fig (8e): Power plot for    Fig(8f): SNR plot for    Fig(8g): Doppler width for    Fig(8f): Noise plot for
East beam  of 22 Jan 2007    East beam  of 22 Jan 2007    East beam  of 22 Jan 2007    East beam of 22 Jan 2007
**Table1 : Average values of SNR, Power and Noise for Each beam of two different data sets**

| Beam | 24 July 2002 | | | | | | 22 Jan 2007 | | | | | |
|---|---|---|---|---|---|---|---|---|---|---|---|---|
| | SNR (dB) | | Power(dB) | | Noise(dB) | | SNR(dB) | | Power(dB) | | Noise(dB) | |
| | FFT | HHT | FFT | HHT | FFT | HHT | FFT | HHT | FFT | HHT | FFT | HHT |
| East | -3.9000 | 1.8708 | 5.3278 | 4.2719 | -17.8649 | -24.0106 | -2.8712 | 2.1041 | 5.1030 | 4.2651 | -18.6521 | -25.1496 |
| West | -3.8707 | 2.0951 | 5.0638 | 4.0296 | -18.1582 | -24.0236 | -2.0108 | 2.3162 | 4.9824 | 3.8526 | -18.2195 | -24.8815 |
| Z-Y | -2.0122 | 3.9113 | 6.3488 | 6.2723 | -18.7317 | -25.4112 | 1.1265 | 5.1198 | 5.8962 | 5.0357 | -19.1154 | -25.0163 |
| Z-X | 1.0270 | 7.1109 | 8.4686 | 7.9878 | -19.6510 | -26.2158 | 1.5381 | 6.2547 | 7.5149 | 6.1491 | -18.3125 | -24.9167 |
| North | -1.4652 | 3.7248 | 6.1634 | 5.3566 | -19.4641 | -25.4609 | -1.3284 | 3.0010 | 5.9632 | 5.1026 | -19.7732 | -25.8064 |
| South | -0.7526 | 4.2647 | 6.6162 | 5.6437 | -19.7239 | -25.7137 | -1.0631 | 3.9962 | 6.1173 | 5.9427 | -19.8451 | -24.5331 |




**Table 2: Average values of SNR, Power and Noise for all the beams**

| 24 July 2002 | | | | | | 22 Jan 2007 | | | | | |
|---|---|---|---|---|---|---|---|---|---|---|---|
| SNR(dB) | | Power(dB) | | Noise(dB) | | SNR(dB) | | Power(dB) | | Noise(dB) | |
| FFT | HHT | FFT | HHT | FFT | HHT | FFT | HHT | FFT | HHT | FFT | HHT |
| -1.8289 | 3.8296 | 6.3314 | 5.5936 | -18.9323 | -25.1393 | -0.7681 | 3.7986 | 5.9295 | 5.0579 | -18.9863 | -25.0506 |

**7. Conclusion**
Thus an efficient algorithm based on Empirical Mode Decomposition de-noising using soft threshold technique
for accurate doppler profile detection and improved SNR for MST Radar Signals was developed. Further,
spectral moments were estimated and signal parameters such as mean doppler, signal power, noise power and
SNR were calculated and the algorithm was tested on different radar data sets for its efficacy in comparison to
FFT. It has been observed that there is a considerable improvement in recognition of the doppler echo and SNR.
**Acknowledgements**
This research work is supported by the Indian Space Research Organization (ISRO under RESPOND scheme)
#B.19012/72/2014-II,12/11/2014. We thank National Atmospheric Research Laboratory (NARL), Gadanki for
providing the MST Radar data and appreciate helpful comments from the Scientists, NARL.

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
