# Peer review of "An Efficient Algorithm for Improved Doppler Profile Detection of MST Radar Signals"

_Geoscientific Instrumentation, Methods and Data Systems, 2017_

## Referee Comment (RC1) · Anonymous Referee #1 · 18 Oct 2017

The authors describe their experiments in using Hilbert Huang Transform in analysing data from atmospheric research radar. I consider this topic to be of interest to the intended audience of the journal and an interesting project in itself. The text in the manuscript is mostly of good quality, although some paragraphs require considerable re-phrasing to be clearer.

However, I have three main issues that need to be seriously addressed and a major revision is needed.

1. I feel that the manuscript reads more like something between a technical report and a scientific article. As the authors are referring to existing earlier work, it is unclear what their actual new contribution is. Are there any similar experiments with this type of data, is this new to the Indian radar etc.?

2. In my opinion, there are too many qualitative terms. For example, there is no justification for calling the algorithm "efficient" unless there is an actual comparison such as the time required to produce the results from raw data using the FFT versus HHT methods. Similarly, why haven't the authors used some, e.g. statistical method for assessing the similarity of the Doppler spectra output from using FFT versus HHT? Where is the support for claiming that the proposed method detects the Doppler more accurately? This is not "clearly visible" in the plots.

3. The methodology includes a number of parameters that the user needs to decide before processing the data and these choices affect the results. For all potential users, the understanding of which selections are critical is of immense value. For credibility, a comprehensive quantitative analysis should be carried out. What the authors have shown is that their method seems to work at least for this particular case. But could one do better? Is this the best or worst case scenario? The authors can – and should – use their expertise in their MST radar data to highlight possible problem areas.

I encourage the authors to dive a little deeper in their experiments as their proposed methods appear promising.

—————

Here are my more detailed remarks:

4. There are a few inconsistencies in the text, so I would recommend going through the manuscript and ensuring all abbreviations are explained when they are introduced for the first time, all units are consistent ("km" not "Km"), EMD algorithm refers to steps (1) and (2) instead of (a) and (b) etc.

5. It would be good to have short paragraph describing what the MST radars are actually used for (and by whom) already in the introduction before discussing the processing techniques. Are the data analysed visually only?

6. In describing the Indian MST radar section, could you insert some real numbers

to provide a general understanding of the normal operations. For example, what does "high resolution" mean, field-of-view etc. Is the Indian radar a typical example of MST radars and what are the major differences? Number of range bins, beams, etc.? Why do you say it is "an excellent system"?

7. Section 3.1, line 58: the sentence "…represented by an IMF that satisfies the conditions of an IMF…" needs to be re-worked.

8. Section 3.1.1 EMD algorithm, lines 68-70: do I understand it correctly that in step (c ), the value of k is used only to keep track on how many times the previous steps were done? And the iterations are stopped when (not "until") the required conditions are met? Please re-phrase.

9. Section 3.1.1 EMD algorithm, line 72: when you talk about the residue $r_n(t)$ becoming less than a predetermined value (see my major point 3), what do you actually mean? The squared sum of the timeseries $r(t)$ or what? I don't also understand what you mean by "monotonic" in this context. Please clarify.

10. Section 3.1.1 EMD algorithm, line 73: "no longer contains any useful frequency information". Please clarify how to quantify this.

11. Section 4 Denoising, lines 89-96: please clarify the whole paragraph, it needs some re-wording.

12. Section 4.2.1 Processing steps (lines 118-136): I think the step (i) and (ii) could be combined into a single step "Read data".

13. Please go through all of the equations, which are not properly typeset. This may be a pdf-conversion issue, but it needs to be checked.

14. Section 6 Results and discussion, lines 158-167: see my major points 2 and 3 and please revise.

15. Section 6, lines 166-177, tables 1-2: You are using four (!) decimal places, such

as 5.6587dB. Do the measurement uncertainties really justify this accuracy? What is a significant difference in SNR given the system noise levels in your radar?

16. Section 6, lines 164-165: I would expect there to be some differences in the "apparent noise" in the data when using either FFT or HHT methods. Did you notice anything, are you able to quantify your observations?

17. Figures 1-10: all plots are much too small to see the details and zooming in the pdf-file does not improve legibility. I would suggest selecting fewer representative plots and possibly highlighting key features (such as "true peaks" based on expert visual analysis) and differences in the outcome. The plots should not be screengrabs. Please revise.

18. References: Please check the journal style guide and go through the references. There were also references to "Donoho et al." and "Donoho and Johnstone" yet there is only a single author "Donoho" in the references. Please check for consistency.

---

## Referee Comment (RC2) · Anonymous Referee #2 · 26 Dec 2017

Paper Title: "An efficient algorithm for improved Doppler profile detection of MST radar signals"

Authors: N. Padmaja, S. Varadarajan, P. Yashoda, and E. Ramyakrishna

Manuscript #: GI-2017-9

General Comments

The manuscript reports on the application of HHT (Hilbert Huang Transformation) to Doppler spectra estimation of atmosphere radar measurements. It introduces a new methodology for potential improvement of Doppler spectra estimation and may potentially be suitable for publication. However, the theory is not described well. The improvement is not specifically described. Thus, the paper needs substantial revision

before publication.

The major concerns are as follows:

Line 67: IFM should be explained more in detail, not only referring to an article, because it one of the essential parts of the method proposed in this paper.

Line 78: No conditions to be satisfied are given in the Step (c).

Line 85: How is it determined whether the residual contains useful information or not?

Line 98: What is the SD value typical to? Does it applies to any signals or some specific cases?

Line 106: Define tau_j, not only referring to articles.

Line 114-117: Equations (3) and (4) are identical.

Line 121: Define "N".

Line 133: Define the ".r" format.

Line 134: Define the ".mat" format.

Line 188: Why can the peaks in HHT results be identified as "true" peaks?

Line 194: Why can the Doppler estimated by HHT be regarded as "accurate"? There is no reference or control data. More validation is needed for improvement of Doppler parameters.

Specific Comments

Line 29: "Kms" should be "km" or "kilometers". Line 30: "echo returns" should be "echoes". Line 40: "Km" should be "km". Line 43: "very high resolution" is ambiguous. How high the resolution is depends on applications. Specific numbers should be given. Line 66: "same" should be "the same". Line 72: "x(t)" should be "X(t)". Line 113: The sentence is incomplete. Line 192: "Table1" should be "Table 1".

Figures: Labels are too small to read. Figure quality is too low to recognize.

---

## Author Comment (AC1) · 6 Feb 2018

Sir,

I thank you for the meticulous review of the paper and for your suggestions.

The Hilbert Huang Transform (HHT) is an empirically based data- analysis method proposed Norden E. Huang in 1996. HHT can be used for processing non-stationary and nonlinear signals. Earlier FFT, wavelets and adaptive moments estimation methods were used and the results were published. This method is applied to Indian MST radar signals for analysis for the first time.

An algorithm was developed to plot the Doppler spectra using both FFT and HHT and further the peak was identified using 3 point moving average method and adaptive moments estimation technique. (Anandan, V. K., et al. "An adaptive moments estimation technique applied to MST radar echoes." Journal of Atmospheric and Oceanic Technology 22.4 (2005): 396-408 We further compared our results with the ADP (Atmospheric Data Processor) software that is already in use at NARL, India to validate for true peak identification and detection. .

We verified for various sets of MST data and validated the results and observed that Doppler estimation using HHT is regarded as more accurate that FFT. As all the results cannot be accommodated in this paper , we have shown the results for two sets of data only. But still we are in the process of validating the results with GPS Radiosonde data that is assumed to be more precise. Presently this work is being carried out.

A typical value for SD is around 0.21 to 0.3. We have taken the value as 0.21 as it is minimum value . But we have tried for other different values also for trial and there was no significant difference in the IMF's.

The amount of resolution depends on the clarity of enabling the algorithm to identify and detect the true peak. It was observed that the developed algorithm is intelligent enough to do the job of identifying the true peak and this was tested for different range bins and a paper was presented at International Union of Radio Science (URSI), 3rd URSI Regional Conference on Radio Science,1 - 4 March 2017 TIRUPATI, India.

Regarding the minor modifications suggested such as kms, rephrasing the sentences and about MST radar systems and data formats and specifications and parameters, references, clarity of figures etc, I would add and modify them suitably.

Thank You Sir.

---

## Author Comment (AC2) · 6 Feb 2018

Sir,

I thank you for the meticulous review of the paper and for your suggestions.

The Hilbert Huang Transform (HHT) is an empirically based data- analysis method proposed Norden E. Huang in 1996. HHT can be used for processing non-stationary and nonlinear signals. Empirical Mode Decomposition (EMD) is direct, adaptive and intuitive, with a posteriori-defined basis, based on the simple assumption that any data consists of simple intrinsic modes of oscillations. Each intrinsic mode function may be, nonlinear or linear, represents a simple oscillation, which will have the same number of extreme and zero-crossings. The oscillation will also be symmetric with respect to the "local mean". At any given time, and the data may have many different coexisting modes of oscillation, one superimposing on the others. Each oscillatory mode is represented by an Intrinsic Mode Functions (IMF) with the following definition.
a) In the whole data base, the number of extrema, and the number of zero crossings must either equal or differ most by one, and
b) At any point, the mean value of the envelope defined by the local maxima and the envelope defined by the local minima is zero.

A typical value for SD is around 0.21 to 0.3. We have taken the value as 0.21 as it is minimum value . But we have tried for other different values also for trial and there was no significant difference in the IMF's.

.r" format refers to the MST Radar RAW data that is stored in the specific format at National Atmospheric Research Laboratory. Similarly ".mat" format refers to the matrix format that is obtained after converting the raw data to". mat" format so that it can be used for processing using MATLAB. The .mat data is in MATRIX form.

$\tau_j$ refers to the universal threshold value. $\Gamma[\tau j]$ is a thresholding function, and $\tau j$ is the threshold parameter, the threshold can be determined in different ways. Donoho and Johnstone proposed a universal threshold, $\tau_j$ for removing noise [13-14]. N is the data size of the Matrix (nxn). As an alternative to minimax threshold Donoho and Johnstone (1994) proposed the universal threshold $\lambda univ = \hat{\sigma} \sqrt{2 \log N}$, where N is the number of pixels and $\hat{\sigma}$ is estimated standard deviation of the noise for an image.

We have initially used three point moving average method to detect the true peak of the echoes and further used the adaptive moments estimation method (Anandan, V. K., et al. "An adaptive moments estimation technique applied to MST radar echoes." Journal of Atmospheric and Oceanic Technology 22.4 (2005): 396-408) for true peak identification and detection. We further compared our results with the ADP (Atmospheric Data Processor ) software that is already in use at NARL, India.

We verified for various sets of MST data and validated the results and observed that Doppler estimation using HHT is regarded as more accurate that FFT. As all the results cannot be

accommodated in this paper , we have shown the results for two sets of data only. But still we are in the process of validating the results with GPS Radiosonde data that is assumed to be more precise. Presently this work is being carried out.

The amount of resolution depends on the clarity of enabling the algorithm to identify and detect the true peak. It was observed that the developed algorithm is intelligent enough to do the job of identifying the true peak and this was tested for different range bins and  a paper was presented at International Union of Radio Science (URSI), 3rd URSI Regional Conference on Radio Science,1 - 4 March 2017 TIRUPATI, India.

 Regarding the minor modifications suggested such as echoes, kms etc I would modify them suitably.

Thank You Sir.